

# Probing the origins of human acetylcholinesterase inhibition via QSAR modeling and molecular docking

Saw Simeon[1,*], Nuttapat Anuwongcharoen[1,*], Watshara Shoombuatong[1], Aijaz Ahmad Malik[1], Virapong Prachayasittikul[2], Jarl E.S. Wikberg[3] and Chanin Nantasenamat[1]

[1] Center of Data Mining and Biomedical Informatics, Faculty of Medical Technology, Mahidol University, Bangkok, Thailand
[2] Department of Clinical Microbiology and Applied Technology, Faculty of Medical Technology, Mahidol University, Bangkok, Thailand
[3] Department of Pharmaceutical Biosciences, Uppsala University, Uppsala, Sweden
[*] These authors contributed equally to this work.

Corresponding author
Chanin Nantasenamat,
chanin.nan@mahidol.ac.th

## ABSTRACT

Alzheimer's disease (AD) is a chronic neurodegenerative disease which leads to the gradual loss of neuronal cells. Several hypotheses for AD exists (e.g., cholinergic, amyloid, tau hypotheses, etc.). As per the cholinergic hypothesis, the deficiency of choline is responsible for AD; therefore, the inhibition of AChE is a lucrative therapeutic strategy for the treatment of AD. Acetylcholinesterase (AChE) is an enzyme that catalyzes the breakdown of the neurotransmitter acetylcholine that is essential for cognition and memory. A large non-redundant data set of 2,570 compounds with reported $IC_{50}$ values against AChE was obtained from ChEMBL and employed in quantitative structure-activity relationship (QSAR) study so as to gain insights on their origin of bioactivity. AChE inhibitors were described by a set of 12 fingerprint descriptors and predictive models were constructed from 100 different data splits using random forest. Generated models afforded $R^2$, $Q^2_{CV}$ and $Q^2_{Ext}$ values in ranges of 0.66–0.93, 0.55–0.79 and 0.56–0.81 for the training set, 10-fold cross-validated set and external set, respectively. The best model built using the substructure count was selected according to the OECD guidelines and it afforded $R^2$, $Q^2_{CV}$ and $Q^2_{Ext}$ values of 0.92 ± 0.01, 0.78 ± 0.06 and 0.78 ± 0.05, respectively. Furthermore, Y-scrambling was applied to evaluate the possibility of chance correlation of the predictive model. Subsequently, a thorough analysis of the substructure fingerprint count was conducted to provide informative insights on the inhibitory activity of AChE inhibitors. Moreover, Kennard–Stone sampling of the actives were applied to select 30 diverse compounds for further molecular docking studies in order to gain structural insights on the origin of AChE inhibition. Site-moiety mapping of compounds from the diversity set revealed three binding anchors encompassing both hydrogen bonding and van der Waals interaction. Molecular docking revealed that compounds **13**, **5** and **28** exhibited the lowest binding energies of −12.2, −12.0 and −12.0 kcal/mol, respectively, against human AChE, which is modulated by hydrogen bonding, $\pi$–$\pi$ stacking and hydrophobic interaction inside the binding pocket. These information may be used as guidelines for the design of novel and robust AChE inhibitors.

# INTRODUCTION

Neurodegenerative diseases is caused by the progressive loss of neural cells thereby leading to nervous system dysfunction (*Beal, 1995*; *Kuca et al., 2016*). In particular, Alzheimer's disease (AD) is a debilitating illness that is expected to triple by the year 2050 (*Brookmeyer et al., 2007*). AD is characterized by gradual cognitive impairment, memory loss and decline in speech, behavioral abnormality and eventually death. The pathological changes in AD are mainly attributed to the dramatic loss of neurons in many areas of the central nervous system accompanied by a great reduction in the levels of neurotransmitters. Acetylcholine (ACh) is a neurotransmitter possessing important cognitive and muscular functions. Particularly, in the peripheral nervous system, ACh is found at the neuromuscular junction where it is involved in muscle contraction while in the central nervous system, it is involved in cognitive functions such as thought, learning and memory.

Acetylcholinesterases (AChE) is an enzyme that catalyzes the breakdown of ACh to choline and acetic acid (*Quinn, 1987*). Thus, a promising therapeutic approach is to maintain the level of ACh by inhibiting the enzyme that is responsible for its breakdown. The structure of AChE is comprised of four main subsites consisting of anionic subsite, esteratic site, oxyanion hole and the acyl pocket (*Bourne, Taylor & Marchot, 1995*). The anionic site is involved in the binding of the positive quaternary amine of ACh (*Ordentlich et al., 1993*). The substrate interacts with the 14 aromatic residues that forms the active site. Of these 14 aromatic residues, Trp84 is important for the enzyme activity because when it is replaced by alanine, the activity of the enzyme decreased by 3,000-fold (*Tougu, 2001*). The esteratic site contains the catalytic triad consisting of Ser203, His447 and Glu334 (i.e., resembling that of chymotrypsin and other serine proteases) that hydrolyzes ACh to acetate and choline (*Harel et al., 1993*). The mechanism of the hydrolysis starts from the carboxyl ester leads to the formation of an acyl-enzyme and choline. Finally, the acyl-enzyme undergoes nucleophilic attack by water molecules thereby regenerating the enzyme (*Tougu, 2001*). The oxyanion hole consisting of Gly121, Gly122 and Ala204 contribute hydrogen bond donors to help stabilize the tetrahedral intermediate of ACh form during catalysis. The acyl pocket consisting of Phe295 and Phe297 are gatekeepers that limit the dimension of substrates that can enter the active site.

AChE inhibitors form one of the most actively investigated classes of compounds having been labeled as a potential agent for the treatment of AD by inhibiting AChE from hydrolyzing ACh, thereby leading to increases in the level of ACh (*Birks, 2006*). Generally, AChE inhibitors can be classified into reversible and irreversible inhibitors. Reversible inhibitor bind to the enzyme at allosteric sites as to reduce the activity of the enzyme whether or not the enzyme has already bind the substrate or not. Tacrine is a reversible AChE inhibitor that was synthesized nearly five decades ago and in 1993 it has become the first drug to be marketed for the treatment of AD with approval from the US. Food and

Drug Administration (*Racchi et al., 2004*). On the other hand, irreversible inhibitors such as metrifonate (*Morris et al., 1998*) bind to the target enzymes and dissociates very slowly from the enzyme via either covalent or non-covalent interactions (*Kitz & Wilson, 1962*).

Quantitative structure–activity relationship (QSAR) is a paradigm that enables the prediction of biological activities for compounds of interest as a function of their descriptors through the use of statistical or machine learning methods (*Nantasenamat et al., 2009*). Aside from the ability to predict the activity, QSAR models have been instrumental in enabling understanding on the origin of these biological activities by means of interpreting the descriptors used in building such models.

Historically, the first QSAR investigation of AChE inhibitors was reported by *Mundy et al. (1978)* almost 40 years ago in which the $\log(1/LD_{50}$ for a series of twelve substituted 0,0-dimethyl 0-(*p*-nitrophenyl) phosphorothioates and 0-analogs was predicted as a function of the octanol/water partition coefficient. Analysis of the literature of QSAR studies of AChE revealed that much of the early studies are classical QSAR models (i.e., Hansch and Free-Wilson approach) that are based on small congeneric compound set and primarily aimed at predicting AChE inhibition as to investigate the toxic effect of pesticides of various chemotypes belonging to either organophosphates (*Mager, 1983*; *Aaviksaar, 1990*) or carbamates (*Su & Lien, 1980*; *Goldblum, Yoshimoto & Hansch, 1981*; *Walters & Hopfinger, 1986*). Recent QSAR studies are based on the use of large and heterogeneous data sets comprising of structurally diverse chemotypes. This include the study from *Yan & Wang (2012)* where they predicted AChE inhibition for a large set of 404 compounds using multiple linear regression and support vector machine. Furthermore, *Lee & Barron (2016)* performed a 3D-QSAR investigation on a large set of 341 compounds comprising of organophosphates and carbamates. Moreover, *Veselinović et al. (2015)* compiled a set of 278 organophosphates for which they developed QSAR models for predicting AChE inhibition using SMILES-based descriptors.

Research in this field had experienced two distinct transitions when viewed from biological and computational viewpoints. Biologically, early QSAR studies treat AChE as a biomarker of toxicity from pesticides while investigations from later years had shifted the focus by viewing AChE as a therapeutic target for the treatment of AD. In regards to the latter point, viewpoint on targeting AChE as a single target for treating AD is starting to be replaced by the multi-target concept in which the treatment for AD can be approached by a panel of key targets (*Fang et al., 2015*; *Huang et al., 2011*). Computationally, early studies are predominantly based on simple 2D-QSAR (*Mundy et al., 1978*; *Su & Lien, 1980*) while later years started to use more sophisticated approach for understanding AChE inhibition encompassing 3D-QSAR (*Deb et al., 2012*; *Lee & Barron, 2016*; *Prado-Prado et al., 2012*), molecular dynamics (*Shen et al., 2002*), molecular docking (*Lu et al., 2011*; *Deb et al., 2012*; *Giacoppo et al., 2015*), pharmacophore modeling (*Lu et al., 2011*; *Gupta & Mohan, 2014*) and statistical molecular design (*Andersson et al., 2014*; *Prado-Prado, Escobar & Garcia-Mera, 2013*).

Herein, we propose the first large-scale QSAR investigation for predicting AChE inhibition, which to the best of our knowledge represents the largest collection of 2,570 non-redundant compounds. QSAR models were built using interpretable learning methods

(e.g., random forest) and descriptors (i.e., molecular fingerprints) as to unravel the underlying AChE inhibitory activity, which was performed in accordance with guidelines of the Organisation for Economic Cooperation and Development (OECD). Molecular docking was also performed on a chemically diverse set of compounds selected from active AChE inhibitors. Together, the ligand and structure-based approach employed in this study is anticipated to be useful in the design and development of robust AChE inhibitors.

## MATERIALS AND METHODS

A summary of the workflow of this study is provided in Fig. 1. Briefly, this included a large-scale QSAR model for predicting and analyzing the AChE inhibition, which was performed in accordance with the OECD guidelines as follows: (i) a data set with a defined endpoint; (ii) an unambiguous learning algorithm; (iii) a defined applicability domain of the QSAR model; (iv) using appropriate measures of goodness-of-fit, robustness and predictivity; (v) a mechanistic interpretation of the QSAR model. Furthermore, molecular docking was also performed on a chemically diverse data set as to elucidate the underlying binding mechanism. To facilitate the reproducibility of the research work performed herein, the data set (Data S1) and codes (Data S2) used to perform the multivariate analysis are provided as Supplemental Information.

### Data set

A data set of inhibitors against human AChE (Target ID CHEMBL220) were compiled from the ChEMBL 20 database (*Gaulton et al., 2012*) that is comprised of a total number of 9,242 bioactivity data points from 5,049 compounds. SMILES notations of the compounds were curated with ChemAxon's Standardizer (*ChemAxon Kft., 2015*) using the same parameter settings as described in our previous study (*Simeon et al., 2016*). The initial data set was assembled from several bioactivity measurement units including (in order of decreasing data size) $IC_{50}$, $K_i$, % activity, % inhibition, MIC, $EC_{50}$, etc. $IC_{50}$ was selected for further investigation as they constituted the largest subset with 4,910 compounds. A closer look revealed that 1,301 compounds had no reported $IC_{50}$ values or had lesser/greater than signs which were subjected to removal thereby leaving 3,609 compounds. Only data points having nM as the bioactivity unit were selected for further study, which produced 3,596 compounds. Furthermore, redundant compounds having different bioactivity values were kept if the standard deviation of $IC_{50}$ was less than 2 and this resulted in 2,571 compounds. Moreover, some compounds were found to have no SMILES notation associated with it and were thus removed. A final data set comprising of 2,570 compounds was obtained.

### Description of inhibitors

AChE inhibitors were encoded by a vector of fingerprint descriptors accounting for its molecular constituents. Prior to calculating descriptors, salts were removed and tautomers were standardized using the built-in function of the PaDEL-Descriptor software (*Yap, 2011*).

Although fingerprint descriptors are able to capture the feature space of chemical compounds, their ability to be used as descriptors for bioactivity modeling can vary. In

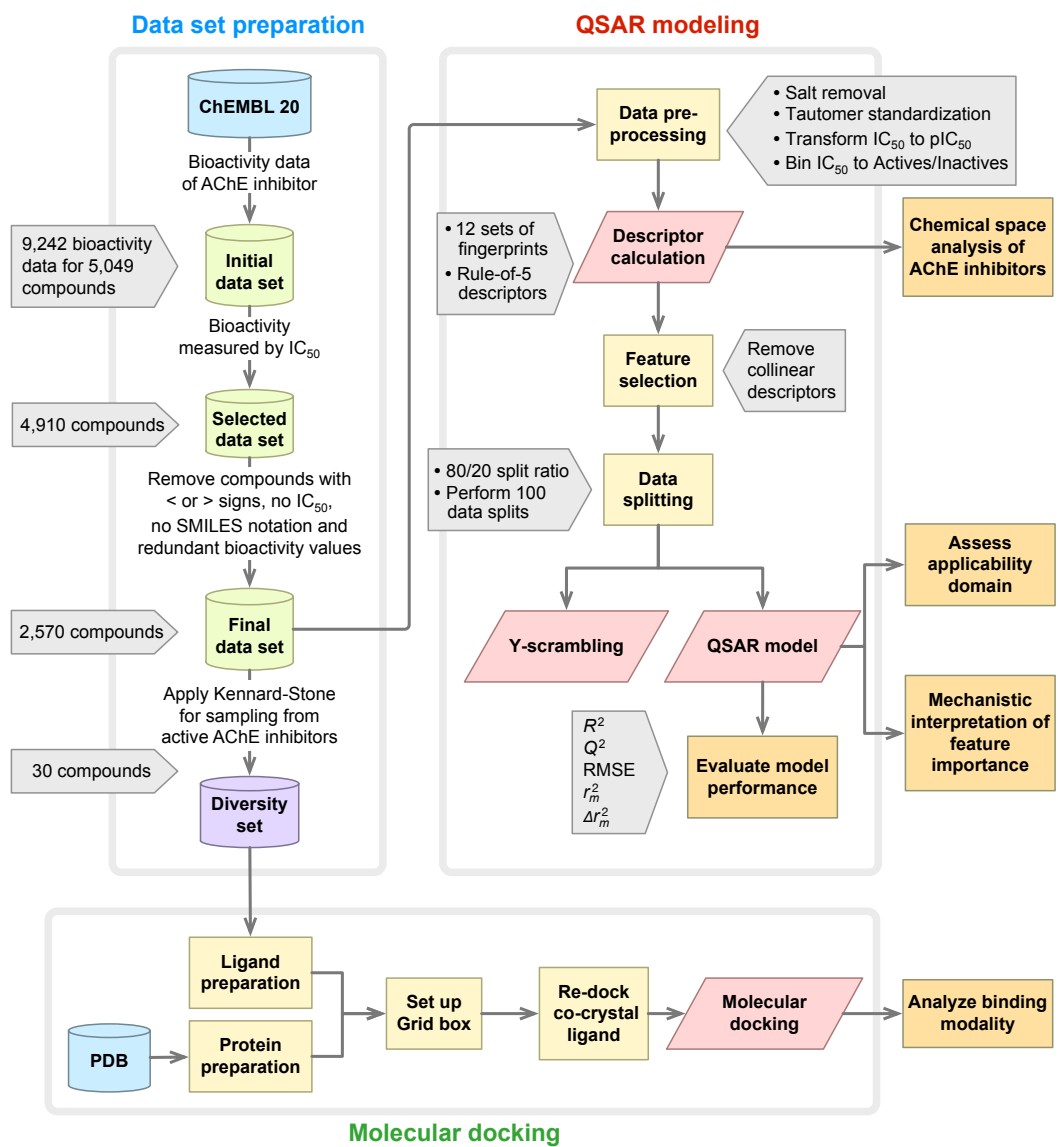

**Figure 1  Workflow of QSAR modeling and molecular docking for investigating AChE inhibitory activity.**

fact, performance differences existing amongst the different fingerprint type has been the subject of several investigations into its utilization for bioactivity modeling. *Riniker & Landrum (2013)* benchmarked and assessed the performance of predictive models constructed from 2D fingerprint descriptors obtained from RDKit.

In this study, the suitability of 12 different fingerprint descriptors for predicting the bioactivity of AChE inhibitors was investigated. Table 1 summarizes the employed fingerprints along with their corresponding size, description and reference.

Additionally, the four molecular descriptors that are used to define the Lipinski's rule-of-five comprising of molecular weight (MW), logarithm of the octanol/water partition coefficient (ALogP), number of hydrogen bond donor (nHBDon) and number of hydrogen bond acceptor (nHBAcc) were also computed by the PaDEL-Descriptor software.

**Table 1  Summary of 12 sets of fingerprint descriptors employed in this study.**

| No. | Fingerprint | Number | Description | Reference |
|-----|-------------|--------|-------------|-----------|
| 1 | CDK | 1024 | Fingerprint of length 1024 and search depth of 8 | *Steinbeck et al. (2003)* |
| 2 | CDK extended | 1024 | Extends the fingerprint with additional bits describing ring features | *Steinbeck et al. (2003)* |
| 3 | CDK graph only | 1024 | A special version that considers only the connectivity and not bond order | *Steinbeck et al. (2003)* |
| 4 | E-state | 79 | Electrotopological state atom types | *Hall & Kier (1995)* |
| 5 | MACCS | 166 | Binary representation of chemical features defined by MACCS keys | *Durant et al. (2002)* |
| 6 | PubChem | 881 | Binary representation of substructures defined by PubChem | *NCBI (2009)* |
| 7 | Substructure | 307 | Presence of SMARTS patterns for functional groups | *Laggner (2005)* |
| 8 | Substructure count | 307 | Count of SMARTS patterns for functional groups | *Laggner (2005)* |
| 9 | Klekota–Roth | 4860 | Presence of chemical substructures | *Klekota & Roth (2008)* |
| 10 | Klekota–Roth count | 4860 | Count of chemical substructures | *Klekota & Roth (2008)* |
| 11 | 2D atom pairs | 780 | Presence of atom pairs at various topological distances | *Carhart, Smith & Venkataraghavan (1985)* |
| 12 | 2D atom pairs count | 780 | Count of atom pairs at various topological distances | *Carhart, Smith & Venkataraghavan (1985)* |

## Feature selection

Collinearity is a condition where descriptor pairs are known to have intercorrelation, which not only add complexity to the model but could potentially give rise to bias. To remedy this, the *cor* function from the R package *stats* was used to find the pairwise correlation among descriptors, and descriptors in a pair with a Pearson's correlation coefficient greater than the threshold of 0.7 was filtered out using the *findCorrelation* function from the R package *caret* to obtain a smaller subset of descriptors (*Kuhn, 2008*).

## Data splitting

To avoid the possibility of bias that may arise from a single data split when building predictive models (*Puzyn et al., 2011*), predictive models were constructed from 100 independent data splits and the mean and standard deviation values of statistical parameters were reported. The data set was split into internal and external sets in which the former comprises 80% whereas the latter constitutes 20% of the initial data set. The *sample* function from the R *base* package was used to split the data.

## Multivariate analysis

Supervised learning is to learn a model from labeled training data which can be used to make prediction about unseen or future data (*James et al., 2013*). This study constructs regression models, which affords the prediction of the continuous response variable (i.e., $pIC_{50}$) as a function of predictors (i.e., fingerprint descriptors).

Random forest (RF) is an ensemble classifier that is composed of several decision trees (*Breiman, 2001*). Briefly, the main idea behind RF is that instead of building a deep decision tree with an ever-growing number of nodes, which may be at risk for overfitting and overtraining of the data, rather multiple trees are generated as to minimize the variance

instead of maximizing the accuracy. As such, the results will be more noisier when compared to a well-trained decision tree, yet these results are usually reliable and robust. The *ranger* function from the R package *ranger*, which is a fast implementation of the RF algorithm that was used for constructing the models (*Wright & Ziegler, 2015*).

## Validation of QSAR models

Model validation is an important process, which should be performed to ensure that a fitted model can accurately predict responses for future or unknown subjects. Two statistical parameters were used to evaluate the performance of the QSAR models consisting of Pearson's correlation coefficient ($r$) and root mean squared error (RMSE). The $r$ value is a commonly used metric to represent the degree of relationship between two variables of interest. It can range from $-1$ to $+1$ in which negative values are indicative of negative correlation between two variables and vice versa. RMSE is a commonly used parameter to assess the relative error of the predictive model. The predictive performance of the QSAR models was verified by 10-fold cross-validation, external validation and Y-scrambling test.

The 10-fold cross-validation technique does not used the entire data set to build predictive model. Instead, it splits the data into training and testing data set by allowing model that is built with training data set us allow to assess the performance of the model on the testing data set. By performing repeats of the 10-fold validation, the average accuracies can be used to truly assess the performance of the predictive model.

Y-scrambling test was used to ensure the robustness of the predictive model not only to rule out the possibility of chance correlations but also to assess the statistical significance of $R^2$ and $Q^2$, ensuring the generalizability of QSAR model. The true Y-dependent variable (i.e., pIC$_{50}$) was randomly scrambled and the statistical assessment parameters are recalculated. Performance of the Y-scrambling test can be deduced from the regression line of the plot of $R^2$ versus $Q^2$. Intercept values for $R^2$ and $Q^2$ as denoted by i$R^2$ and i$Q^2$, respectively, were calculated. Negative i$Q^2$ is indicative of an acceptable QSAR model and that there is no chance correlation from the real model (*Eriksson et al., 2003*). Furthermore, $r_m^2$ and $\Delta r_m^2$ metrics as introduced by *Roy et al. (2013)* were used to verify the robustness of the proposed QSAR model in which an acceptable QSAR model should give $r_m^2 > 0.5$ and $\Delta r_m^2 < 0.2$.

## Applicability domain analysis

The applicability domain (AD) estimates the likelihood of reliable prediction for compounds on the basis of how similar they are to compounds used to build the model. Thus, compounds falling outside the AD may lead to unreliable predictions. The most common approach for determining AD is described by *Gramatica (2007)* and *Tropsha, Gramatica & Gombar (2003)*, which is to compute the leverage values for each compound. The leverage value allows one to identify whether new compounds will lie within or outside the domain. Leverage values for all compounds are calculated via adjustment of $X$ to give the hat matrix $H$:

$$H = X(X^T X)^{-1} X^T \tag{1}$$

where $X$ is a two-dimensional matrix comprising of $n$ compounds and $m$ descriptors while $X^T$ is the transpose of $X$. Meanwhile, the leverage value of the $i$th compound ($h_i$) is the $i$th diagonal element of $H$:

$$h_i = x_i^T (X^T X)^{-1} x_i \qquad (2)$$

where $x_i$ is the descriptor row-vector of the $i$th compound. The warning leverage $h^*$ is calculated by:

$$h^* = 3(p+1)/n. \qquad (3)$$

Practically, the leverage value along with the William's plot is often used to assess the AD of QSAR models. The William's plot is constructed by depicting the standardized residuals versus the leverage value for each compound's $h_i$. If the $i$th compound has $h_i > h^*$ then it means that the $i$th compound exerts a great influence on the QSAR model and may be excluded from the AD. In spite of this, it does not appear to be an outlier because its standardized residual may be small.

## Molecular docking

The co-crystal structure of human AChE with donepezil (PDB ID: 4EY7) was retrieved from the Protein Data Bank and initially prepared by removing alternative side chains and water molecules. The protein was prepared via the rebuilding of bonds and the addition of missing hydrogen atoms. Subsequently, the protein was cleaned by merging the atomic charge and removing lone pair atoms, non-polar hydrogen atoms and non-standard amino acid residues. Grid box was set up to provide coverage of the active site of human AChE with a dimension of $40 \times 30 \times 40$ Å ($X$, $Y$ and $Z$ axes of $-13.987$, $-41.668$ and $27.109$, respectively). Molecular docking was consequently performed with AutoDock Vina (*Trott & Olson, 2010*) using default parameters. The docking protocol was validated in order to ensure its reliability for subsequent analysis of the studied compounds. This was performed by extracting the co-crystal ligand, donepezil, from the PDB file and re-docked to the co-crystal human AChE protein. The root mean squared deviation (RMSD) of the atomic position between the original orientation of the co-crystal ligand and the re-docked ligand is computed and is deemed acceptable if the RMSD value is less than or equal to 2.0 Å.

A set of 30 representative and chemically diverse compounds, which will be referred hereafter as the diversity set, were extracted from the full set of active AChE inhibitors (i.e., $IC_{50} < 1$ µM) using the Kennard–Stone algorithm (*Kennard & Stone, 1969*). These compounds were used as ligands for molecular docking against the human AChE. The binding energy (kcal/mol) of AChE inhibitors were calculated according to the built-in scoring function of Autodock Vina and conformers providing the lowest binding energy were selected for further analysis of the binding mode. Furthermore, key-interacting residues and their moiety preferences were analyzed using LigPlot+ (*Wallace, Laskowski & Thornton, 1995*), Maestro (*Schrödinger, 2015b*) and the SiMMap web server (*Bollback, 2006*). Finally, three-dimensional structure of protein–ligand interaction was created and visualized using Pymol (*Schrödinger, 2015a*).
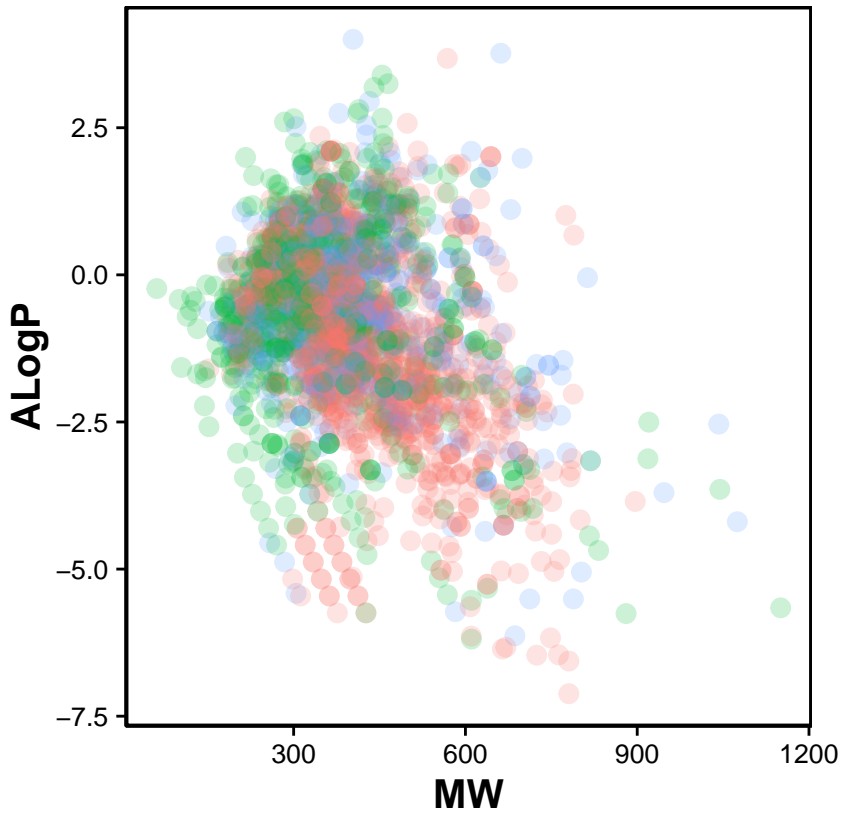

**Figure 2** **Chemical space of AChE inhibitors.** Actives, inactives and intermediates are shown in green, red and blue colors, respectively.

## RESULTS AND DISCUSSION

### Chemical space of AChE inhibitors

Navigation of the chemical space of AChE inhibitors was performed to gain insights into the structure–activity relationship by analyzing the Lipinski's rule-of-five descriptors. Chemical space analysis may provide important knowledge on the general character of compounds governing inhibitory properties of compounds. Exploratory data analysis was performed using the Lipinski's rule-of-five descriptors comprising of MW, ALogP, nHBDon and nHBAcc. MW represents the molecular size of a compound that is commonly used because of it can be easily interpreted and calculated as well as appropriate size of a compound is important for its passage via lipid membrane. ALogP is a widely used parameter for determining the lipophilicity of a compound and used in calculating the membrane penetration and permeability of compounds. nHBDon and nHBAcc describe the number of hydrogen bond donors and hydrogen bond acceptors, respectively, which is used to measuring hydrogen bonding capacity. Visualization of the chemical space of ALogP as a function of MW is shown in Fig. 2, as to investigate the chemical space of AChE inhibitors. A dense distribution of inhibitors was observed within the space of MW starting from approximately 300–600 Da and within the space of ALogP ranging from approximately −2.5 to 2.5. In addition, the box plot of the Lipinski's descriptors is

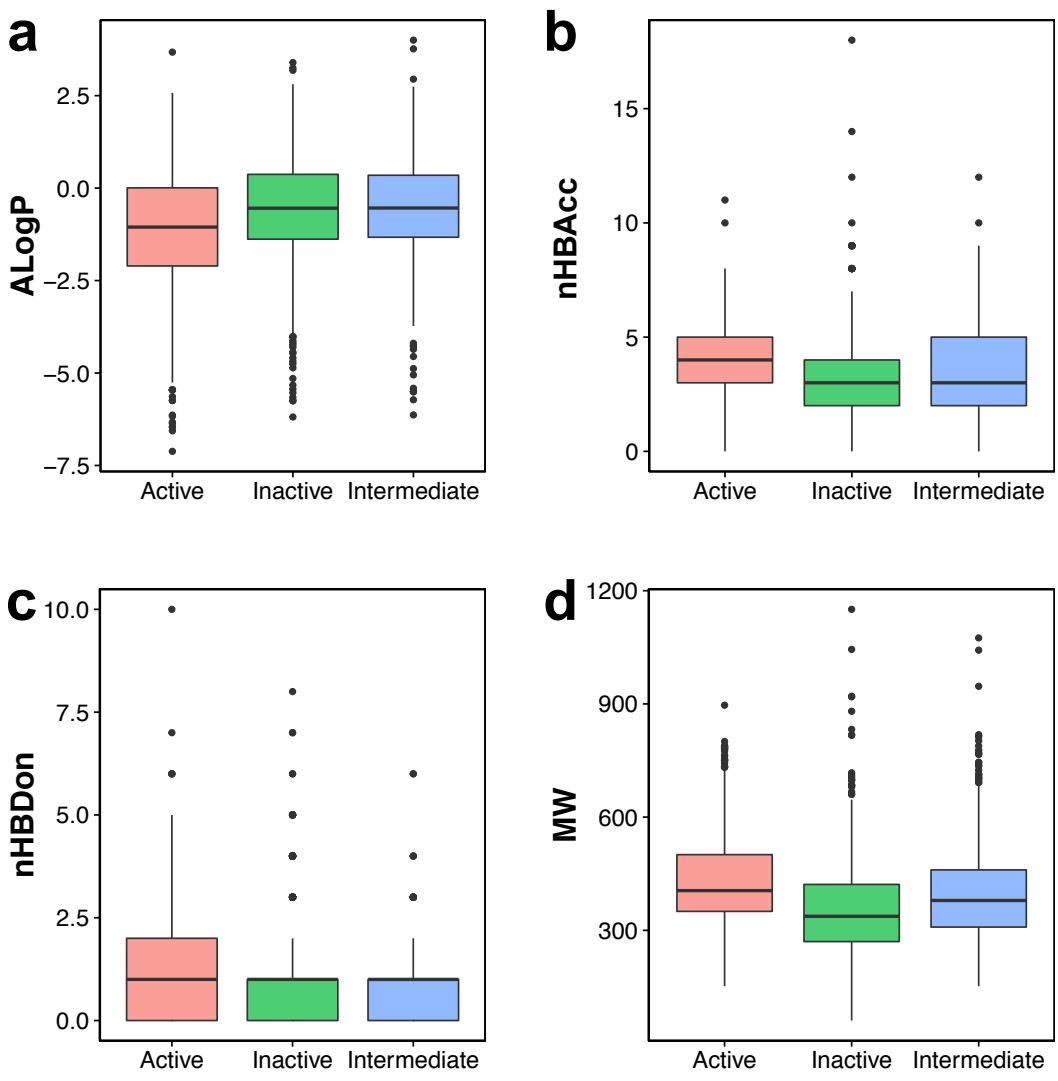

**Figure 3**   Box plot of AChE inhibitors using Lipinski's rule-of-five descriptors.

shown in Fig. 3. Compounds with negative ALogP values approximately of closer to 0.0 can be found in inactive inhibitors whereas most of the active inhibitors tend to possess approximately lower values in average of ALogP values.

Visual representation of the overall distribution of data values of Lipinski's descriptors is shown as box plots in Fig. 3 in which the ALogP, nHBAcc, nHBDon and MW are shown in Figs. 3A, 3B, 3C and 3D, respectively. Analysis of the box plots revealed that there were no differences amongst the three bioactivity classes for nHBAcc and nHBAcc as deduced from the boundaries of the boxes (i.e., representing the first and third quartiles). ALogP and MW were found to display differences amongst the bioactivity classes. Particularly, ALogP values for actives were the lowest while negligible differences were observed for the other two classes. Furthermore, MW for actives were the largest amongst the three bioactivity classes, which is followed by the intermediates while inactives were smallest.

## QSAR model for predicting AChE inhibitory activity

A data set comprising of 2,570 compounds were used for construction of QSAR models. Particularly, twelve sets of fingerprint descriptors were benchmarked in order to find the best performing set. Prior to modeling, feature selection was applied to remove collinear descriptors. Each of the twelve models were then built using a data split ratio of 80/20 in which 80% of the data set was used as the internal set and 20% as the external set. This procedure was iteratively performed in which each of the 100 independent data splits were used for model construction and the performance results given in Table 2 are the mean and standard deviation values derived from these runs.

It can be observed that all twelve models are capable of capturing the inhibitory activity space of AChE inhibitors as they provided $R^2$ and $Q^2$ (i.e., both 10-fold CV and external sets) greater than the threshold values proposed by *Golbraikh & Tropsha (2002)* of 0.6 and 0.5, respectively, which is indicative of robust model performance. The possibility of chance correlation can be assessed from the $R^2$–$Q^2$ margin as described by *Eriksson & Johansson (1996)* in which values <0.2–0.3 are indicative of predictive and reliable models while values >0.2–0.3 suggests possible chance correlation or the presence of outliers in the data set. Furthermore, observation of the $Q^2_{CV}$–$Q^2_{Ext}$ margin revealed that the difference was negligible with values in the range of 0 and 0.01.

Generally, it can be seen that models with larger descriptor size, namely CDK and CDK extended, afforded the best performance with $Q^2_{CV}$ of 0.79 ± 0.07 and 0.79 ± 0.06, respectively, and $Q^2_{Ext}$ of 0.80 ± 0.04 and 0.81 ± 0.04, respectively. The opposite also holds true as the model with the least number of descriptors were also found to perform the worst amongst the other fingerprints with $Q^2_{CV}$ of 0.55 ± 0.09 and $Q^2_{Ext}$ of 0.56 ± 0.05. In a nutshell, the model performance in order of decreasing value is as follows: CDK extended > CDK > MACCS ≈ Substructure count ≈ Klekota–Rota count > PubChem > Klekota–Roth ≈ 2D atom pairs count > CDK graph only > Substructure > 2D atom pairs > E-state.

The best performing model is not necessarily the best choice considering the fact that the descriptor size for the best models were quite high and is consequently prone to overfitting. It was found that the substructure count provided reasonably good predictive performance (i.e., $Q^2_{CV}$ and $Q^2_{Ext}$ of 0.78 ± 0.06 and 0.78 ± 0.05, respectively) with the advantage of making use of a small set of 26 descriptors. Therefore, this fingerprint was selected for further interpretation of the feature importance.

To further check the reliability and validity of the selected model, Y-scrambling test was performed for 100 iterations. Table 3 demonstrates that QSAR models built using substructure count has a low $Q^2$ (−0.0013), which rules out the possibility of chance correlation. Furthermore, model afforded an $r^2_m$ value of 0.61 ± 0.06 thereby revealing its robustness. It is observed in Table 3 that the value of $\Delta r^2_m$ is greater that 0.2 but also close to 0.20.

As shown in Fig. 4, it can also be seen that scatter plots of experimental versus predicted pIC$_{50}$ of panels A, C and F displayed narrower variance of the data points than the other methods as assessed via 10-fold cross-validation and external set.

Simeon et al. (2016), *PeerJ*, DOI 10.7717/peerj.2322

**Table 2** **Performance summary of QSAR models for predicting pIC$_{50}$.**

| Descriptor class | N | Training set | | 10–fold CV set | | External set | | $R^2 - Q^2_{CV}$ | $R^2 - Q^2_{Ext}$ |
|---|---|---|---|---|---|---|---|---|---|
| | | $R^2$ | RMSE$_{Tr}$ | $Q^2_{CV}$ | RMSE$_{CV}$ | $Q^2_{Ext}$ | RMSE$_{Ext}$ | | |
| CDK | 960 | 0.93 ± 0.01 | 0.44 ± 0.04 | 0.79 ± 0.07 | 0.76 ± 0.15 | 0.80 ± 0.04 | 0.73 ± 0.09 | 0.14 | 0.13 |
| CDK extended | 948 | 0.94 ± 0.01 | 0.42 ± 0.03 | 0.79 ± 0.06 | 0.76 ± 0.12 | 0.81 ± 0.04 | 0.72 ± 0.08 | 0.15 | 0.13 |
| CDK graph only | 198 | 0.87 ± 0.01 | 0.61 ± 0.03 | 0.72 ± 0.06 | 0.87 ± 0.13 | 0.72 ± 0.05 | 0.87 ± 0.09 | 0.15 | 0.15 |
| E-State | 21 | 0.66 ± 0.03 | 1.00 ± 0.05 | 0.55 ± 0.09 | 1.11 ± 0.13 | 0.56 ± 0.05 | 1.10 ± 0.08 | 0.11 | 0.10 |
| MACCS | 77 | 0.89 ± 0.01 | 0.56 ± 0.03 | 0.77 ± 0.07 | 0.81 ± 0.15 | 0.77 ± 0.04 | 0.80 ± 0.09 | 0.12 | 0.12 |
| PubChem | 103 | 0.90 ± 0.01 | 0.55 ± 0.03 | 0.76 ± 0.05 | 0.80 ± 0.11 | 0.78 ± 0.03 | 0.79 ± 0.08 | 0.14 | 0.12 |
| Substructure | 30 | 0.75 ± 0.01 | 0.85 ± 0.03 | 0.64 ± 0.06 | 1.00 ± 0.13 | 0.65 ± 0.05 | 0.98 ± 0.08 | 0.11 | 0.10 |
| Substructure count | 26 | 0.92 ± 0.01 | 0.50 ± 0.02 | 0.78 ± 0.06 | 0.77 ± 0.14 | 0.78 ± 0.05 | 0.77 ± 0.10 | 0.14 | 0.14 |
| Klekota–Roth | 111 | 0.89 ± 0.01 | 0.59 ± 0.03 | 0.74 ± 0.07 | 0.85 ± 0.14 | 0.76 ± 0.05 | 0.81 ± 0.10 | 0.15 | 0.13 |
| Klekota–Roth count | 72 | 0.91 ± 0.01 | 0.52 ± 0.03 | 0.78 ± 0.06 | 0.79 ± 0.14 | 0.78 ± 0.05 | 0.77 ± 0.11 | 0.13 | 0.13 |
| 2D atom pairs | 42 | 0.75 ± 0.03 | 0.85 ± 0.06 | 0.61 ± 0.08 | 1.03 ± 0.15 | 0.60 ± 0.06 | 1.05 ± 0.12 | 0.14 | 0.15 |
| 2D atom pairs count | 38 | 0.92 ± 0.01 | 0.51 ± 0.02 | 0.74 ± 0.07 | 0.84 ± 0.15 | 0.76 ± 0.05 | 0.82 ± 0.10 | 0.18 | 0.16 |

Simeon et al. (2016), *PeerJ*, DOI 10.7717/peerj.2322

**Table 3  Performance summary of QSAR models assessed using additional statistical metrics.**

| Descriptor class | N | Training set | | 10-fold CV set | | External set | | iR² | iQ² |
|---|---|---|---|---|---|---|---|---|---|
| | | $r_m^2$ | $\Delta r_m^2$ | $r_m^2$ | $\Delta r_m^2$ | $r_m^2$ | $\Delta r_m^2$ | | |
| CDK | 960 | 0.82 ± 0.02 | 0.07 ± 0.01 | 0.62 ± 0.09 | 0.20 ± 0.06 | 0.64 ± 0.05 | 0.19 ± 0.03 | 0.0003 | −0.0003 |
| CDK extended | 948 | 0.83 ± 0.01 | 0.07 ± 0.01 | 0.62 ± 0.08 | 0.20 ± 0.05 | 0.65 ± 0.05 | 0.18 ± 0.03 | 0.0006 | −0.0005 |
| CDK graph only | 198 | 0.70 ± 0.02 | 0.14 ± 0.01 | 0.51 ± 0.07 | 0.27 ± 0.05 | 0.53 ± 0.05 | 0.26 ± 0.03 | 0.0007 | −0.0006 |
| E-State | 21 | 0.35 ± 0.03 | 0.38 ± 0.02 | 0.27 ± 0.07 | 0.40 ± 0.04 | 0.28 ± 0.05 | 0.41 ± 0.03 | 0.0011 | −0.0009 |
| MACCS | 77 | 0.73 ± 0.01 | 0.12 ± 0.01 | 0.57 ± 0.09 | 0.23 ± 0.05 | 0.58 ± 0.05 | 0.23 ± 0.03 | 0.0005 | −0.0004 |
| PubChem | 103 | 0.74 ± 0.02 | 0.12 ± 0.01 | 0.57 ± 0.07 | 0.23 ± 0.04 | 0.59 ± 0.05 | 0.22 ± 0.03 | 0.0006 | −0.0005 |
| Substructure | 30 | 0.50 ± 0.02 | 0.28 ± 0.01 | 0.39 ± 0.07 | 0.34 ± 0.05 | 0.41 ± 0.05 | 0.33 ± 0.03 | 0.0033 | −0.0027 |
| Substructure count | 26 | 0.77 ± 0.01 | 0.10 ± 0.01 | 0.60 ± 0.08 | 0.22 ± 0.05 | 0.61 ± 0.06 | 0.21 ± 0.04 | 0.0015 | −0.0013 |
| Klekota–Roth | 111 | 0.71 ± 0.02 | 0.14 ± 0.01 | 0.54 ± 0.08 | 0.25 ± 0.05 | 0.56 ± 0.06 | 0.24 ± 0.03 | 0.0006 | −0.0004 |
| Klekota–Roth count | 72 | 0.76 ± 0.02 | 0.11 ± 0.01 | 0.60 ± 0.08 | 0.22 ± 0.05 | 0.61 ± 0.07 | 0.21 ± 0.04 | 0.0006 | −0.0005 |
| 2D atom pairs | 42 | 0.49 ± 0.03 | 0.28 ± 0.02 | 0.35 ± 0.08 | 0.36 ± 0.04 | 0.35 ± 0.06 | 0.36 ± 0.03 | 0.0010 | −0.0008 |
| 2D atom pairs count | 38 | 0.75 ± 0.01 | 0.10 ± 0.01 | 0.52 ± 0.05 | 0.26 ± 0.05 | 0.54 ± 0.06 | 0.25 ± 0.04 | 0.0006 | −0.0005 |

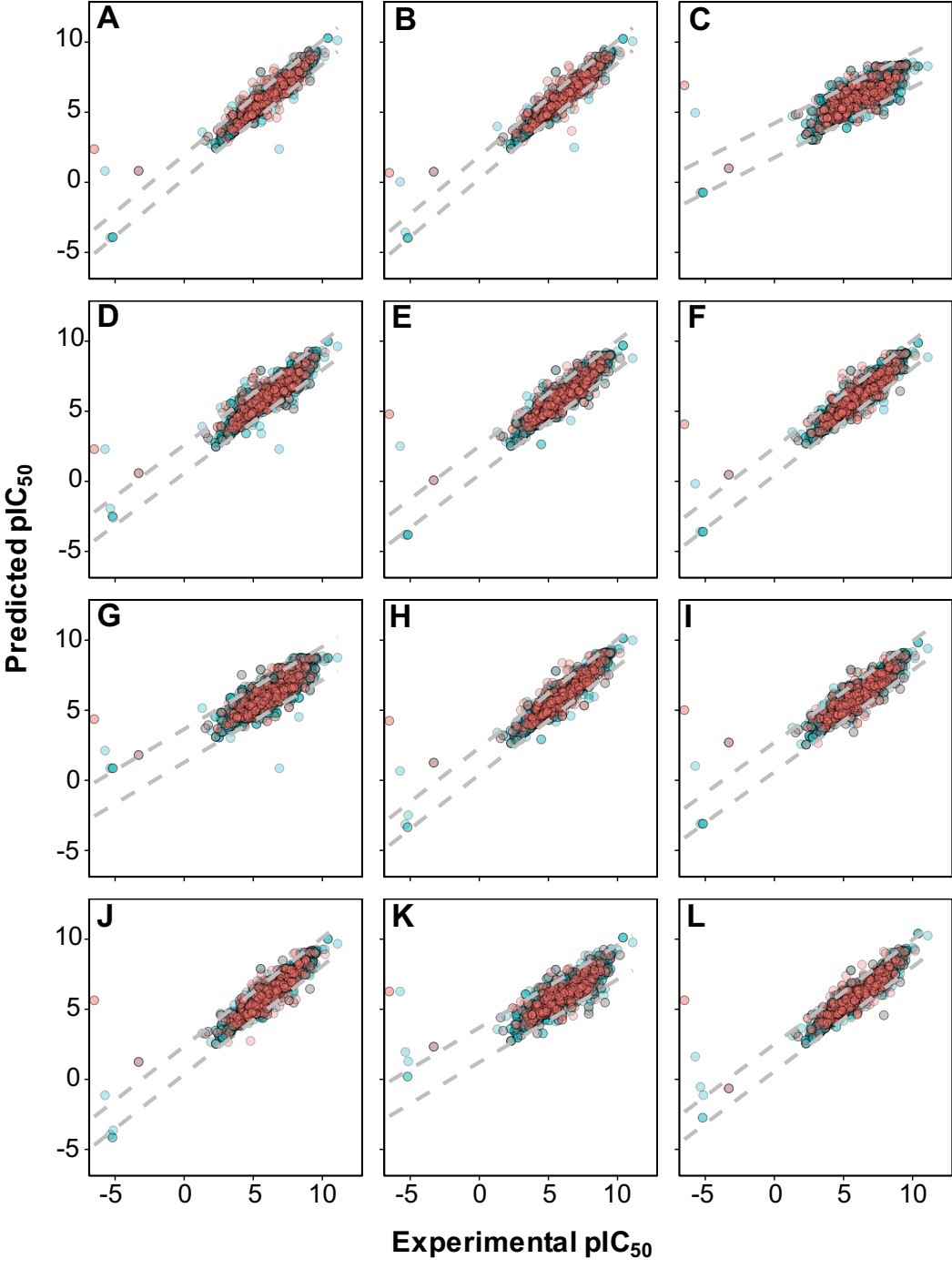

**Figure 4** **Plot of experimental versus predicted pIC$_{50}$ values for models constructed with 12 different fingerprint descriptors.** Shown are models built with CDK fingerprint (A), CDK extended fingerprint (B), E-State fingerprint (C), CDK graph only fingerprint (D), MACCS fingerprint (E), PubChem fingerprint (F), substructure fingerprint (G), substructure fingerprint count (H), Klekota–Roth fingerprint (I), Klekota–Roth fingerprint count (J), 2D atom pairs (K) and 2D atom pairs count (L).

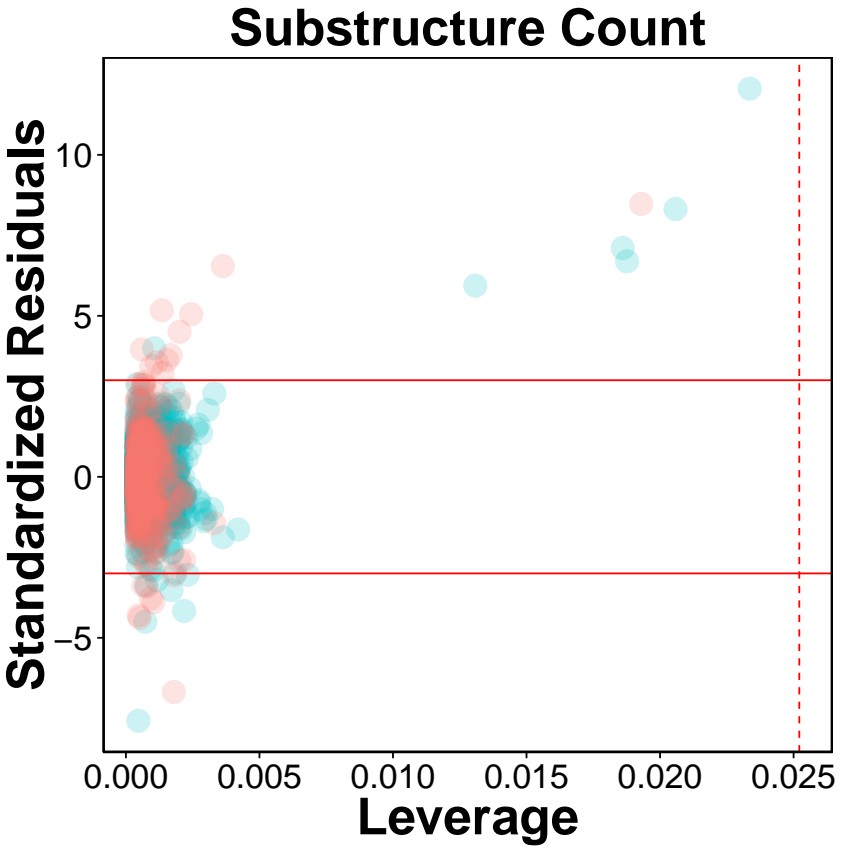

**Figure 5** **Evaluation of applicability domain using William plots for QSAR model built with substructure fingerprint count.** Compounds in the internal and external sets are shown by blue and red dots, respectively. The solid and dashed lines correspond to the ±3 standardized residual and the warning leverage value ($h^* = 0.025$), respectively.

## Applicability domain

The AD of the proposed QSAR model was defined as provided by the Williams plot shown in Fig. 5. The employed data set consisting of 2,570 compounds was randomly split to two separate subset in which the first subset constituting 80% of the data set was used as an internal set while the second subset constituting the remaining 20% were used as an external set. Compounds representing the internal set (blue dots) and external set (red dots) are shown in the Williams plot and it can be clearly seen that almost all of the 2,570 compounds were located within the boundaries of applicability domain, which indicated that our proposed QSAR model had a well-defined AD.

As can be seen in Fig. 5, very few compounds indeed fall outside the ±3 standardized residual range. This consisted of six compounds (**997, 1829, 62, 1096, 13, 677**) from the internal set and 11 compounds (**2116, 2120, 2388, 2117, 2392, 2323, 2423, 2424, 2507, 2219, 2422**) from the external set that had standardized residual higher than 3. On the other hand, 7 compounds (**1567, 576, 1644, 1098, 2022, 1447** and **322**) from the internal set and 8 compounds (**2486, 2353, 2130, 2553, 2389, 2072, 2103, 2125**) from the external

set had standardized residual lower than $-3$. The corresponding chemical structures are provided in Table S1.

## Mechanistic interpretation of feature importance

Feature importance analysis help reveal features that are important toward bioactivity. There are essentially two parameters for evaluating the relative importance of features used in models using the RF algorithm: (i) accuracy and (ii) Gini index (i.e., variance of the responses). The latter was selected as a metric for ranking important features (i.e., mean decrease of the Gini index) for predicting the pIC$_{50}$ of AChE inhibitors (Fig. 6). Table 4 lists the substructure fingerprints along with their respective descriptions.

As can be seen in Fig. 6, the top ranking feature is secondary carbon (SubFPC2), which is a carbon atom with two carbon neighbors. In the context of drug design, such central carbon atom may be more difficult to be accessed and metabolized by cytochrome P450 (*Uetrecht & Trager, 2007*) and therefore are more metabolically stable.

The second most important feature is the rotatable bond (SubFPC302). Based on the rule of three for defining lead-like compounds, a compound may have a lead-like characters if it does have rotatable bonds of no more than 3. On the other hand, *Veber et al. (2002)* noted that the the upper limit of a orally bioavailable drugs is of seven rotatable bonds. Nevertheless, it has been found that number of rotatable bonds provide better discrimination between compounds that are orally active and those that are not. *Kryger, Silman & Sussman (1999)* claimed that E2020 (i.e., also known as donepezil and marketed as Aricept) needs at least two rotatable bonds on each side of the piperidine in which two aromatic moieties of E2020 interact with Trp86 and Trp286 (human AChE numbering), suggesting that links between aromatic systems of the inhibitor against its AChE counterparts are essential to yield high affinity.

The third important substructure is the aromatic ring (SubFPC274). Findings from X-ray crystallographic study showed that in the binding site of the co-crystal structure of AChE with tacrine, the aromatic ring of acridine engages in a $\pi-\pi$ stacking interaction with the indole of Trp86 (human AChE numbering), thereby indicating the importance of the aromatic ring for AChE inhibition (*Chen et al. 2012*).

The fourth important feature is C ONS bond (SubFPC295), which is defined as the presence of any carbon connected with either oxygen, nitrogen or sulfur atom in a molecule. These atoms are considered as high electron density atoms, which exerted from higher electronegativity comparing with a carbon atom. Unequal sharing of electron pair making covalent bond contribute polarity and afford dipole moment to a molecule, which able to generate a dipole–dipole attraction such as hydrogen bond between two polar molecules. Furthermore, increasing of polarity and presenting of hydrogen bond improve water solubility, which is essential characteristic of a drug.

The fifth important substructure is secondary mixed amine (SubFPC32). The importance of the moiety was demonstrated in the work of *Bembenek et al. (2008)* in which a structure-based approach was used to reveal that in the catalytic triad, Trp86 interacts with the quaternary amine of ACh through a cation–$\pi$ interaction. Furthermore, in the

**Table 4  List of top substructure fingerprints and their corresponding description.**

| Fingerprints | Description |
| --- | --- |
| SubFPC1 | Primary carbon |
| SubFPC2 | Secondary carbon |
| SubFPC3 | Tertiary carbon |
| SubFPC5 | Alkene |
| SubFPC18 | Alkylaryether |
| SubFPC23 | Amine |
| SubFPC26 | Tertiary aliphatic amine |
| SubFPC28 | Primary aromatic amine |
| SubFPC32 | Secondary mixed amine |
| SubFPC35 | Ammonium |
| SubFPC49 | Ketone |
| SubFPC88 | Carboxylic acid derivative |
| SubFPC100 | Secondary amide |
| SubFPC135 | Carbonyl derivative |
| SubFPC137 | Vinylogous ester |
| SubFPC143 | Carbonic acid diester |
| SubFPC153 | Urethan |
| SubFPC171 | Arylchloride |
| SubFPC180 | Hetero N basic H |
| SubFPC181 | Hetero N nonbasic |
| SubFPC182 | Hetero O |
| SubFPC184 | Heteroaromatic ring |
| SubFPC274 | Aromatic ring |
| SubFPC275 | Heterocyclic ring |
| SubFPC276 | Epoxide |
| SubFPC287 | Conjugated double bond |
| SubFPC295 | C ONS bond |
| SubFPC296 | Charged |
| SubFPC298 | Cation |
| SubFPC300 | 1,3-Tautomerizable |
| SubFPC301 | 1,5-Tautomerizable |
| SubFPC302 | Rotatable bond |
| SubFPC303 | Michael acceptor |
| SubFPC307 | Chiral center specified |

'anionic' site Trp286 appears to attract the amine moiety via cation and/or hydrophobic interactions.

The sixth, seventh and eighth important substructures are heterocyclic rings (SubFPC275), tertiary carbon (SubFPC3) and primary carbon (SubFPC1). Heterocycles are of high relevance in the design of AChE inhibitors as it allows $\pi-\pi$ stacking interaction with key amino acid residues in the binding site of AChE. It is observed that the binding site of the AChE are highly hydrophobic in nature. Particularly, the Trp286 (human AChE numbering) which is a part of the peripheral anionic site of the AChE is involved in the

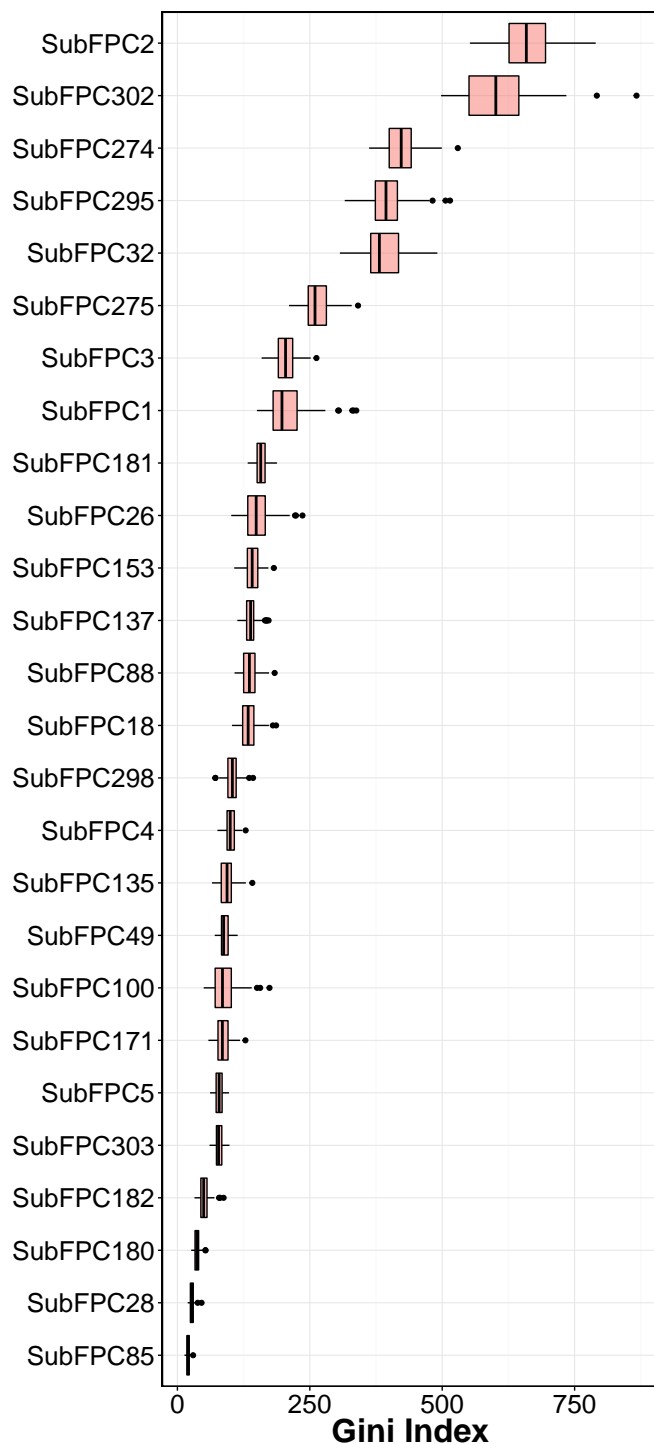

**Figure 6** Plot of feature importance as exemplified by the Gini index.

$\pi-\pi$ interaction with heterocycles of AChE inhibitors (*Lu et al., 2011*). The aforementioned explanation made for the secondary carbon is also applicable for the tertiary carbon in which the higher number of carbon neighbors would also confer high stability against cytochrome P450 metabolism.

The ninth and tenth important substructures are Hetero N nonbasic (SubFPC181) and tertiary aliphatic amine (SubFPC26) in which the former is defined as aromatic nitrogen. The presence of the nitrogen atom, which is a cationic moiety deemed to interact with aromatic residues through $\pi$-cation interaction as observed in E2020 against *Torpedo californica* AChE (tcAChE). The charge of nitrogen atom located in piperidine ring provide $\pi$-cation interaction with the side chain of Phe337 (Phe330 in tcAChE numbering) (*Guo et al., 2004*). Since the active site gorge of AChE is comprised of several aromatic residues known as the aromatic patch, the addition of cationic moiety could possibly increase the binding affinity when the ligand is arranged in a suitable conformation with respect to the aromatic side chain of residues in the active site.

It is worthy to note that substructures pertaining to the covalent inhibitors, carbamates and organophosphates, were not found in the top ten important features. A manual inspection of the 307 substructure fingerprints revealed that there were none specifically describing the characteristic feature of carbamate and organophosphates. However, there were quite a few substructures that resembled partial features of the carbamate (e.g., carboxylic acid (SubFPC84), carboxylic ester (SubFPC85), "NOS methylen ester and similar" (SubFPC65), etc.) as well as substructures resembling partial features of organophosphates (e.g., phosphonic monoester (SubFPC230), phosphonic diester (SubFPC231), phosphonic monoamide (SubFPC232), phosphonic esteramide (SubFPC234), phosphonic acid derivative (SubFPC235), etc.). In spite of the presence of these descriptors, important features obtained from predictive model as revealed by the Gini index did not contain these features in the top ten. A possible explanation for such observation could be that there were a few carbamates (123 compounds) and organophosphates (18 compounds) present in the data set and its relative importance may have been masked by other features that represented the other structural class. Thus, it seems very lucrative for a future large-scale study to be performed focusing on these two compound classes.

## MOLECULAR DOCKING OF AChE INHIBITORS

To gain a further understanding on the non-covalent interaction between AChE and their inhibitors, a chemically diverse set of 30 representative compounds was extracted from active AChE inhibitors (i.e., having $IC_{50}$ <1 $\mu$M) using the Kennard–Stone algorithm and subjected to an investigation on its binding modality against the active site of AChE. Figure 7 shows the distribution of the selected subset of compounds in the context of the full set of actives, which was found to provide a full coverage of the original chemical space. The chemical structures of these compounds are shown in Fig. 8.

The active site of this enzyme is buried inside a narrow gorge of 20 Å deep, which permits multiple enzyme-substrate interaction thereby facilitating the formation of the transition state of ACh (*Silman & Sussman, 2008*; *Zhou, Wang & Zhang, 2010*; *Cheung et*
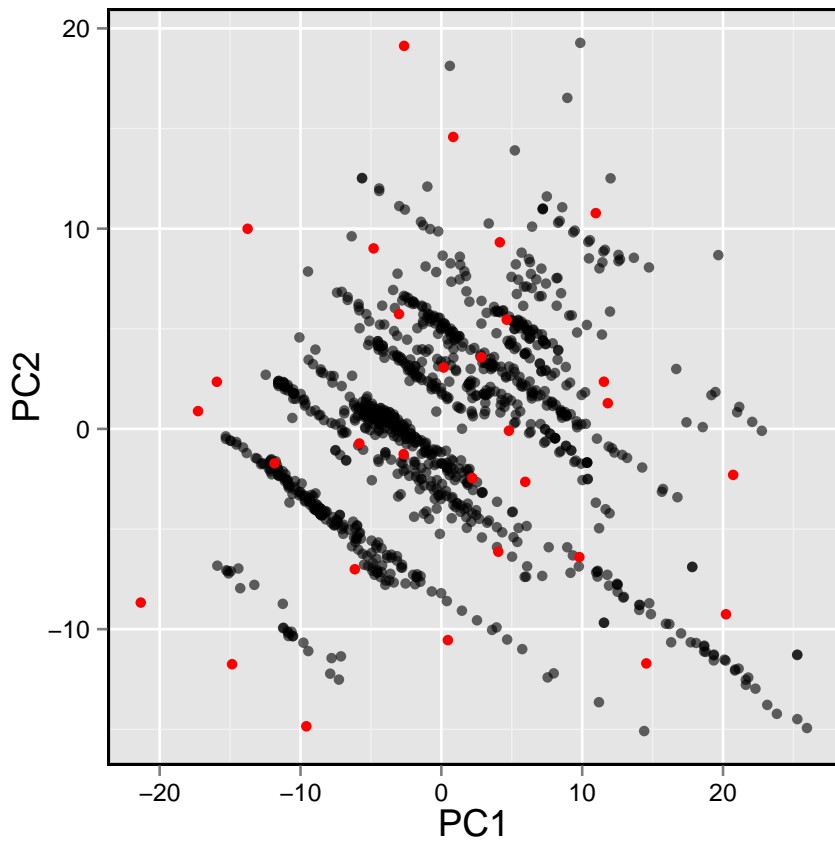

**Figure 7** Plot showing the distribution of active AChE inhibitors (gray circles) and the diversity set (red circles) selected for molecular docking.

*al., 2012*). The entry of the active site gorge is lined up by peripheral anionic site (PAS), which is composed of Tyr72, Tyr124, Trp286 and Try341. The function of PAS is to trap the substrate via $\pi$-cation interaction and proceed through the constriction residues Tyr124 and Phe338 and onto the catalytic site (*Silman & Sussman, 2008*; *Dvir et al., 2010*). As a serine hydrolase, AChE contains Ser203, Glu334 and His447 in the catalytic triad that catalyzed the acylation and deacylation of ACh. The catalytic triad is surrounded by the catalytic anionic site (CAS) (i.e., contains Trp86, Glu202 and Tyr337), oxyanion hole (i.e., comprising of Glu121, Glu122 and Ala204) and the acyl pocket (i.e., comprised of Phe295 and Phe297). These sites interact with ACh and positions it in a suitable orientation for interacting with the catalytic triad as well as providing proton transfer that is essential for nucleophilic substitution during the catalytic reaction (*Zhou, Wang & Zhang, 2010*). As a result of the acylation process, the proton transfer from Ser203 to His447 induces the oxygen atom of Ser203 to engage in the nucleophilic attack of the carbonyl group of ACh, which consequently breaks down the choline moiety and forms a covalent acylenzyme complex between Ser203 and the acetyl group. This complex consequently proceeds with deacylation, which follows a similar mechanism with the acylation stage. The protonation of Glu202 provides a water molecule and the proton transfer from His447 to the water molecule leads to the nucleophilic attack against the

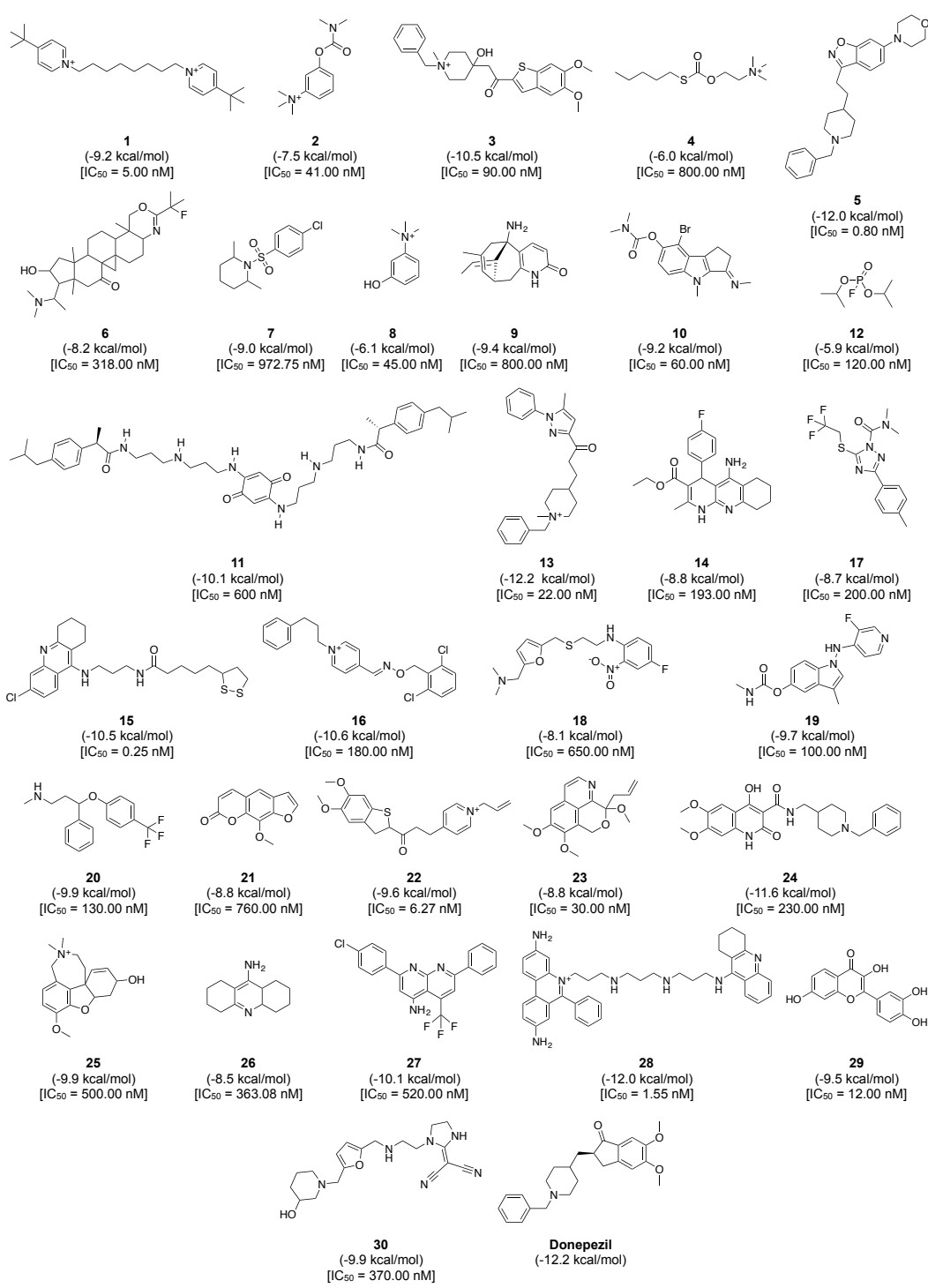

**Figure 8** Chemical structures, binding energy and bioactivity of the diversity set consisting of 30 representative compounds from active AChE inhibitors.

acetyl group of the complex. Finally, this results in the breaking down of the complex thereby restoring wild-type AChE and causing the release of acetic acid from the active site (*Zhou, Wang & Zhang, 2010*). In the case of organophosphate and carbamate poisoning, the active site of AChE is phosphorylated or carbamylated, which is no longer capable to hydrolyze the ACh substrate. Organophosphates target and phosphorylates Ser203 in the same manner as that of ACh. However, the phosphoryl moiety is highly stable and leads to the irreversible inhibition of AChE. The mechanism of carbamate inhibition is virtually identical to organophosphate poisoning with the exception that the carbamylated serine moiety is less stable and is therefore able to be regenerated to the active enzyme form. Such understanding on the catalytic mechanism of substrate and inhibitory mechanism of AChE provides useful insights for the development of therapeutic agents targeting AChE (*Fukuto, 1990*).

Prior to carrying out the molecular docking calculations, the docking protocol was validated by re-docking the co-crystal ligand and protein. It was found that the re-docked ligand exhibited negligible deviation from the co-crystal conformation with an RMSD value of 0.963 Å, which was deemed to be suitable for further molecular docking investigation and its subsequent interpretation. Consequently, the binding modality was analyzed in order to gain understanding on the contribution of key residues in interacting with the investigated set of 30 compounds. This was performed using the SiMMap web server, which revealed three major binding anchors: Hbond1, vdW1 and vdW2 along with their site-moiety preferences. The first anchor site involves hydrogen bond interaction between Tyr124 (i.e., an important residue in the PAS that is spatially located as a bottleneck between the peripheral region and the catalytic site of AChE) and the following ligand moieties: secondary amide, secondary amine, nitrogen moiety in aromatic ring, ketone and ester. Such interaction can be observed in the co-crystal structure of huperzine A with human AChE (*Cheung et al., 2012*). Interestingly, analysis of the important features from QSAR models also revealed the importance of "C ONS bond," "secondary mixed amine," "heterocyclic" and "hetero N non-basic" as they were found to be in the top ten important substructures and is therefore crucial for forming hydrogen bonds. Furthermore, the other anchor sites involve van der Waals interaction in which members of the first van der Waals interaction site (vdW1) are comprised of Tyr124, Phe338 and Tyr341, which has a preference to interact with heterocyclic, aromatic, phenol and other non-polar moieties from representative inhibitors. The second van der Waals' interaction site (vdW2) is consisted of Trp86 and Gly121 with preference for the following ligand moieties: aromatic ring, heterocyclic ring, aliphatic moiety with alkene linkage and phenol moiety. These residues contain either bulky aromatic ring or non-polar moiety as their side chain to provide the van der Waal's surface contact against non-polar moiety from ligands. Notably, aromatic and heterocyclic substructures were also observed in the top ten important substructures for predicting the inhibitory activity of human AChE inhibitors.

Furthermore, analysis of the binding energy from the 30 representative compounds revealed that compounds **13**, **5** and **28** exhibited the lowest binding energy of −12.2, −12.0 and −12.0 kcal/mol, respectively, when interacting with the human AChE binding site, which is comparable to donepezil (−12.2 kcal/mol) as indicated in Fig. 8. Key interacting

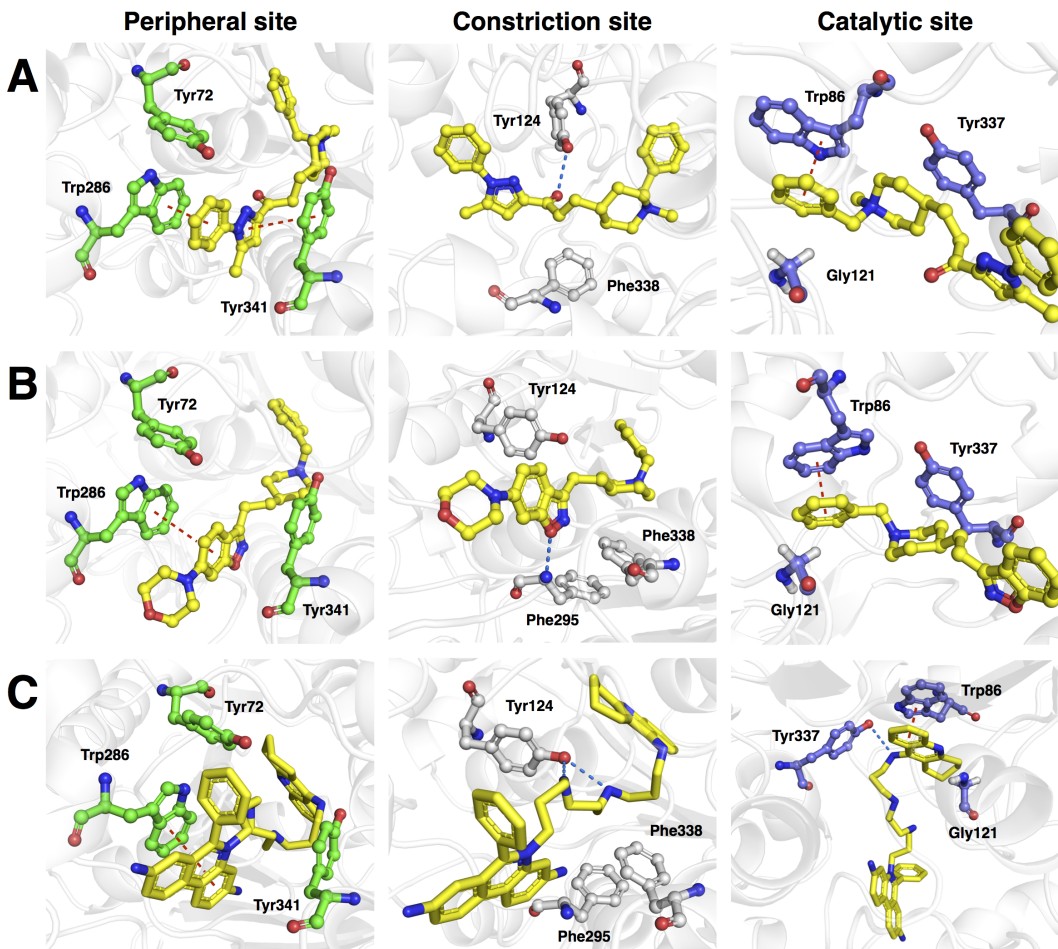

**Figure 9** **Molecular docking of the top three representative compounds against AChE.** The binding be-tween three subsites of AChE with **13**, **5** and **28** are shown in (A, B) and (C), respectively. The representative compounds are highlighted as yellow sticks while residues participating in the peripheral site, constriction site and catalytic site are labeled as green, white and blue sticks, respectively. Hydrogen bonds are shown by blue dash line while the red dash line represents the $\pi-\pi$ stacking interaction between ligand and residues in the active site of AChE.

residues and their moiety preference was deduced from the protein–ligand interaction diagram generated by LigPlot+ (*Wallace, Laskowski & Thornton, 1995*) in combination with Maestro (*Schrödinger, 2015b*) and their three-dimensional structure was visualized by PyMOL (*Schrödinger, 2015a*).

Figure 9A revealed three major interaction subsites for **13** inside the binding pocket of human AChE. The first interaction site (Fig. 9A) is formed by residues from the PAS at the gorge opening consisting of Trp286 and Tyr341 both of which engages in $\pi-\pi$ stacking interaction where the terminal benzene attached with pyrazole moiety of the ligand interacts with the former residue while the pyrazole moiety of **13** provide contact with the phenol moiety of the latter residue, which are deemed to increase the binding fitness against the active site of human AChE. The second interaction subsite (Fig. 9A) is dominated by Tyr124 and Phe338 in which the former interacts with the ketone moiety
of **13** by means of a hydrogen bond, which tends to increase the binding affinity of this compound. The side chain of Phe338 is involved in hydrophobic interaction with carbon atoms from the piperidine ring and the central aliphatic chain of **13**. It can be noted that these residues are members of constriction site, which arranged as bottleneck of active site. The third interaction subsite (Fig. 9A) is moderated by $\pi-\pi$ stacking between Trp86 and the terminal benzene with an attached piperidine moiety as well as hydrophobic interaction between Tyr337 and the piperidine moiety of **13** in which both residues belonged to the CAS. In addition, Gly121 of oxyanion hole also interact with terminal benzene of the ligand through hydrophobic contact increasing binding affinity against catalytic site of AChE.

Analysis of the binding modality of compound **5** revealed interactions with all subsites of the AChE active site gorge as illustrated in Fig. 9B. PAS was the first subsite dominated by $\pi$-$\pi$ stacking between Trp286 and the benzisoxazole moiety of compound **5**, which is essential for stabilizing the binding affinity of the ligand against entry into the gorge. The second interaction site was observed at the constriction site in which the piperidine moiety makes contact with Phe338 via hydrophobic interaction thereby increasing the binding fitness against the bottleneck region of the active site. Similar hydrophobic interaction was also observed in the binding pocket in which Tyr337 from CAS interacts with the piperidine moiety and Gly121 of the oxyanion hole interacts with the terminal benzene of compound **5**. The $\pi$-$\pi$ interaction between the terminal benzene and Trp86 from the CAS was deemed to increase the binding fitness with the catalytic site of AChE. In addition, hydrogen bond interaction is facilitated by the nitrogen atom from Phe295 at the acyl pocket to the oxygen atom from the benzisoxazole moiety. Notably, all sites from the active site gorge are snugly bound by compound **5**, which is deemed to exhibit strong intermolecular interaction with human AChE.

The binding energy of compound **28** was similar to that of compound **5** as indicated in Fig. 8. These compounds possessed several aromatic rings at both terminal, which are favourable for interacting with aromatic residues lining up the surface of the gorge and these are known as the aromatic patch. The 5,7-dihydrophenanthridine moiety facilitates $\pi-\pi$ stacking with the side chain of Trp286 from PAS at the gorge opening. Meanwhile, this moiety also engages in hydrophobic interaction with Phe295, which tends to increase the binding fitness for the acyl pocket. Aside from the former moiety, 1,2,3,4-tetrahydroacridine at the opposite terminal provides $\pi-\pi$ interaction with Trp86 and hydrophobic contact with Tyr337 where both of which are members of CAS in the catalytic site. Furthermore, long aliphatic chain linking the two aromatic moieties provide hydrophobic contact with several aromatic residues in the aromatic patch consisting of Tyr72, Tyr124, Trp286, Tyr337, Phe338 and Tyr341 (i.e., these residues are the members of PAS, CAS and constriction site of the gorge). Moreover, this chain contain several nitrogen atoms, which can act as hydrogen bond donor to Tyr124 and Tyr337 from PAS and CAS, respectively. This would tighten the binding between compound **28** and the active site gorge. The binding modality of this compound was shown in Fig. 9C.

It should be noted that compounds exhibiting strong binding fitness against AChE are those that interact with residues from both PAS and CAS at the entry and inner pocket of the gorge, respectively, as dual-binding site inhibitor through either $\pi-\pi$ stacking or

$\pi$-cation interaction together with hydrophobic contact. These compounds competes with the natural substrate in interacting with these residues. For non-covalent inhibitors, the aromatic moiety is preferred for occupying the interaction sites while hydrophobic moieties are preferred for making contact with the aromatic residues surrounding the catalytic site. Hydrogen bond donors such as secondary amine and heterocyclic ring can be employed for interacting with the oxygen atom on the side chain of Tyr residues. Interestingly, this finding is corroborated by the feature importance results obtained from the QSAR model as shown in Fig. 6 in which the aromatic moiety, C ONS bond, secondary mixed amine, heterocyclic ring and the hetero N non-basic moiety were found amongst the top ten important substructures that are essential for the bioactivity of AChE inhibitor.

## CONCLUSION

In conclusion, twelve sets of fingerprint descriptors were used for constructing QSAR models and their performances were comparatively evaluated. It was observed that several fingerprint descriptors afforded good performance for the constructed models indicating that they could capture the feature space of AChE inhibitors. By taking advantage of the built-in feature importance estimator from RF known as the Gini index, the following important features that are critical for AChE inhibition were identified: secondary carbon (SubFPC2), rotatable bond (SubFPC302), aromatic (SubFPC274), C ONS bond (SubFPC295), secondary mixed amine (SubFPC32) and heterocyclic (SubFPC275). Results from molecular docking also support the aforementioned findings from the QSAR models in which the aromatic, heteroaromatic and heterocyclic rings were preferable moieties for interacting with the hydrophobic pocket of AChE. It is anticipated that the knowledge gained from this study could be used as general guidelines for the design of novel AChE inhibitors.

## ACKNOWLEDGEMENTS

The authors would like to thank the peer reviewers and academic editor for providing comments that helped to improve this manuscript.

### Funding

This research is supported by a Research Career Development Grant (No. RSA5780031) and a New Scholar Grant (No. MRG5980220) from the Thailand Research Fund to CN and WS, respectively, as well as the Swedish Research Links program (No. C0610701) from the Swedish Research Council to CN and JESW. The funders had no role in study design, data collection and analysis, decision to publish, or preparation of the manuscript.

## Grant Disclosures

The following grant information was disclosed by the authors:

Thailand Research Fund: RSA5780031, MRG5980220.

Swedish Research Council: C0610701.

## Competing Interests

The authors declare there are no competing interests.

## Author Contributions

- Saw Simeon, Nuttapat Anuwongcharoen, Watshara Shoombuatong and Aijaz Ahmad Malik performed the experiments, analyzed the data, wrote the paper, prepared figures and/or tables, reviewed drafts of the paper.
- Virapong Prachayasittikul and Jarl E.S. Wikberg analyzed the data, contributed reagents/materials/analysis tools, wrote the paper, reviewed drafts of the paper.
- Chanin Nantasenamat conceived and designed the experiments, performed the experiments, analyzed the data, contributed reagents/materials/analysis tools, wrote the paper, prepared figures and/or tables, reviewed drafts of the paper.

## Data Availability

The data set has been supplied as Supplemental Dataset.

## Supplemental Information

Supplemental information for this article can be found online at http://dx.doi.org/10.7717/peerj.2322#supplemental-information.

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
