# Peer review of "Probing the origins of human acetylcholinesterase inhibition via QSAR modeling and molecular docking"

_PeerJ, doi:10.7717/peerj.2322_

## Round 0.1 · original submission · Minor Revisions

· Academic Editor

Minor Revisions

Dear authors,

The manuscript has been evaluated by three reviewers. They suggested only minor corrections. Please see reviewers comments and submit the revised version.

Sincerely yours,

Walter.

·

Basic reporting

The currnet manuscripts is in the scope of the journal , however, the work seems to be carefully done, some few points need attention before publication.

Experimental design

In the present contribution, the authors proposed a large-scale QSAR investigation for predicting AChE inhibition.

Validity of the findings

I believe that the idea is original and the work could contribute to the development of new AChE inhibitors.

Comments for the author

Whereas the work seems to be carefully done, some few points need attention before publication. In my view the idea is original and the work could contribute to the development of new AChE inhibitors. The QSAR study presented seems good to me. In this line, please find bellow some suggestions to improve the quality of the manuscript.

1) The authors should report applicability domain of the developed models according to Chemom Intell Lab Sys, 145, 2015, 22-29, http://dx.doi.org/10.1016/j.chemolab.2015.04.013.

2) The authors should also use rm2 metrics for validation. See J Comput Chem 34, 2013, 1071-1082, http://dx.doi.org/10.1002/jcc.23231 and Journal of Chemistry, vol. 2016, Article ID 9198582, 12 pages, 2016. doi:10.1155/2016/9198582. Please introduce those papers.

3) A deeper discussion about the docking results is need.

4) Finally, the following references could be introduced in the paper:

JOURNAL OF BIOMOLECULAR STRUCTURE & DYNAMICS. VOL.: 33, 2048-2058 (2015).

CURRENT BIOINFORMATICS VOL. 8, 496-510 (2013).

JOURNAL OF THE BRAZILIAN CHEMICAL SOCIETY, VOL. 27, 641-649 (2016).

CURRENT TOPICS IN MEDICINAL CHEMISTRY VOL. 12, 1843-1865 (2012).

Reviewer 2 ·

Basic reporting

Overall, the introduction explained well about background. However, it is needed to explain more about inhibition mechanisms of AChE. For example, carbamates and organophosphates inhibit AChE by forming covalent bond with catalytic serine residue, and dual binding site inhibitors for reversible binding and block the access of ACh.

Experimental design

Materials and methods section includes sufficient information about data set development and model design. However, there is no concern about mechanism of actions of different inhibitors. I guess some of the selected substructure fingerprints represent important interaction features of inhibitors for forming the covalent bond. For example, C=O in carbamate and P=O in organophosphate are reaction center for forming the covlent bond

Validity of the findings

The selected fingerprints and observed interactions from docking study are reasonable.

Comments for the author

This work dose not provide new information about mechanism of action or interaction features for AChE inhibitor. However, their model is based on the largest data set and provides reasonable performance.

Reviewer 3 ·

Basic reporting

No comments

Experimental design

No Comment

Validity of the findings

No comment

Comments for the author

The work has been carried out with care. The reviewer feels that this manuscript can be accepted. However, a few minor points to be addressed.

In abstract section " line 4 should be " This is not the only reason for AD. There are many hypothesis for AD (Cholinergic, Amyloid, Tau hypothesis etc). As per cholinergic hypothesis, this may be one of the reason of AD.

In Fig. 2 and 3 color code has been changed. Similar code should be followed for uniformity.

---

## Round 0.2 · accepted · Accept

· Academic Editor

Accept

The manuscript has improved a great deal and it can be accepted for publications as it is.